# Community Mobility and COVID-19 Dynamics in Jakarta, Indonesia

**DOI:** 10.3390/ijerph19116671

**Published:** 2022-05-30

**Authors:** Ratih Oktri Nanda, Aldilas Achmad Nursetyo, Aditya Lia Ramadona, Muhammad Ali Imron, Anis Fuad, Althaf Setyawan, Riris Andono Ahmad

**Affiliations:** 1Center for Tropical Medicine, Faculty of Medicine, Public Health and Nursing, Universitas Gadjah Mada, Yogyakarta 55281, Indonesia; oktrinanda1910@gmail.com (R.O.N.); mail.aldilas@gmail.com (A.A.N.); anisfuad@ugm.ac.id (A.F.); 2Department of Health Behavior, Environment and Social Medicine, Faculty of Medicine, Public Health and Nursing, Universitas Gadjah Mada, Yogyakarta 55281, Indonesia; alramadona@ugm.ac.id; 3Wildlife Laboratory, Faculty of Forestry, Universitas Gadjah Mada, Yogyakarta 55281, Indonesia; maimron@ugm.ac.id; 4Department of Biostatistics, Epidemiology, Population Health, Faculty of Medicine, Public Health and Nursing, Universitas Gadjah Mada, Yogyakarta 55281, Indonesia; 5Department of Reproductive Health, Faculty of Medicine, Public Health and Nursing, Universitas Gadjah Mada, Yogyakarta 55281, Indonesia; althaf@ugm.ac.id

**Keywords:** COVID-19, community mobility, Jakarta, statistical modelling, mobility

## Abstract

In response to the COVID-19 pandemic, mobile-phone data on population movement became publicly available, including Google Community Mobility Reports (CMR). This study explored the utilization of mobility data to predict COVID-19 dynamics in Jakarta, Indonesia. We acquired aggregated and anonymized mobility data sets from 15 February to 31 December 2020. Three statistical models were explored: Poisson Regression Generalized Linear Model (GLM), Negative Binomial Regression GLM, and Multiple Linear Regression (MLR). Due to multicollinearity, three categories were reduced into one single index using Principal Component Analysis (PCA)**.** Multiple Linear Regression with variable adjustments using PCA was the best-fit model, explaining 52% of COVID-19 cases in Jakarta (R-Square: 0.52; *p* < 0.05). This study found that different types of mobility were significant predictors for COVID-19 cases and have different levels of impact on COVID-19 dynamics in Jakarta, with the highest observed in “grocery and pharmacy” (4.12%). This study demonstrates the practicality of using CMR data to help policymakers in decision making and policy formulation, especially when there are limited data available, and can be used to improve health system readiness by anticipating case surge, such as in the places with a high potential for transmission risk and during seasonal events.

## 1. Introduction

The novel coronavirus, further called SARS-CoV-2, first emerged in November 2019 in Wuhan, China. The epidemic subsequently began in December 2019, and the cases increased in January 2020. On 30 January, the World Health Organization (WHO) declared the pandemic a “Public Health Emergency of International Concern.” Around eight months after the cases were first identified in China, the pandemic has not yet been contained and has impacted more than 200 countries across continents [1].

Human mobility could act as a vector for the spread of infectious agents. Therefore, mobility restriction is a common strategy to slow down the pandemic. Some countries, such as China and New Zealand, have already demonstrated the success of epidemic containment by putting strict social distancing measures as the priority in the early stage of the epidemic [2,3]. However, different policies in some countries and some other factors contributed to the longer duration of the pandemic in several countries, such as Indonesia. 

A previous study of the H1N1 epidemic reported that population mobility could be a good predictor for the epidemic spread in international travel [4]. In response to the COVID-19 pandemic, digital technology usage, such as mobile application innovation, has been leveraged for disease tracking by people movement to mitigate the infection spread [5]. Furthermore, various mobile-phone data on population movement were made publicly available. In the early stage of the COVID-19 pandemic, a study noted the potential of assessing mobility data in building epidemiological models to anticipate COVID-19 spread [6]. Previous studies demonstrated that building statistical/mathematical modelling from mobility data sources would help to explain the contribution of mobility to COVID-19 cases [7,8].

Google Community Mobility Reports (CMR) is a mobility data source that features population movement patterns in six categories: retail and recreation, groceries and pharmacy, transit stations, parks, workplaces, and residential. These reported mobility data reflect the implementation of mobility restrictions and other events, such as seasonal/public holidays. Several studies have utilized CMR data during the COVID-19 pandemic in cross-country comparisons [9], national levels [10] and sub-national levels [11]. CMR data have also been utilized to explore mobility changes in relation to the implementation of government policy of restrictions at the social-economic level [12], and to investigate how prevention policies, such as the closure of public places, impacted human mobility behavior in different mobility categories. A previous study indicated that stay-at-home requirements, and school and public transport closure reduced human mobility in workplaces, retail, transit, and grocery shopping [5].

When some countries implemented strict lockdown measures, Indonesia did not enforce a national lockdown policy. Instead, partial mobility restrictions, known as large-scale social restrictions (PSBB), were implemented considering local epidemiological status. The provincial government of the Greater Jakarta Area implemented different mobility restrictions from early March until November 2020. As Indonesia’s capital city and the epicenter of the early COVID-19 epidemic, the regulation mainly emphasized limiting population movements by postponing public events, limiting transportation (MRT, LRT, and bus) routes and their capacity. The government also shut down schools and other public places that might cause large gatherings and put into effect a work-from-home policy in specific fields [13].

The changes in mobility restriction regulations were subsequently followed by mobility relaxations (PSBB Transisi) at several periods. This relaxation, along with events (long weekends or annual religious holidays), might contribute to increased population movement within the city and lead to a COVID-19 case surge. By the end of 2020, COVID-19 in Jakarta has exceeded 180,000 cases [14]. Despite the increase in cases, less is known about how much impact population movement has on COVID-19 cases in Jakarta and its further implications for local regulation. Therefore, this study aimed to explore whether human mobility data could be used to predict COVID-19 dynamics in Jakarta.

## 2. Materials and Methods

### 2.1. Data on Cases of COVID-19

We analyzed longitudinal data of COVID-19 daily confirmed cases in Jakarta provided by Kawal COVID-19 website from the period of 2 March (the first two reported cases in Jakarta) to 31 December 2020. Kawal COVID-19 is an independent website that provides information and collect data on Indonesia’s COVID-19 pandemic from the government at the provincial levels. Hence, the data are more reliable [14].

### 2.2. Data on Community Mobility

We retrieved mobility data from the Google Community Mobility Report (CMR), where Google collected anonymized users’ location history data [15]. We used daily aggregated mobility data from 15 February to 31 December 2020. The data show the changes in visits to places categorized as retail and recreation, groceries and pharmacies, parks, transit stations, workplaces, and residential. The information is collected by considering the number of requests made to Google Maps for directions in several countries, sub-regions, cities, and minimum thresholds for direction requests per day. The report then shows the percentage of mobility change in retail and recreation, groceries and pharmacy, parks, workplaces, transit stations, and residential. The descriptions of the places are as follows:-Retail and Recreational: mobility towards places such as restaurants, cafes, shopping centers, museums, libraries, and picture theatres;-Grocery and Pharmacy: mobility trends for places such as grocery shops, food warehouses, markets, local hats, farmer’s markets, specialty food shops, different drug or medicine stores, and pharmacies;-Parks: places of attraction including local parks, national parks, public beaches, marinas, dog parks, plazas, and public gardens;-Transit stations: a process by which a person moves from one place to places like public transport hubs such as subway, bus, and train stations;-Workplaces: the process of going to places of work from a home;-Residential mobility: mobility in the direction of places of residence where a person lived.

We applied statistical transformations to our data and used STATA software 14.0, StataCorp LP, 4905 Lakeway Drive, College Station, TX, USA) to analyze the data. We included the five CMR categories (retail and recreation, groceries and pharmacies, parks, transit stations, and workplaces) as the study exposures and the observed new COVID-19 daily confirmed cases as the study outcome. We excluded residentials because they do not indicate increased mobility [9,16]. There is some limitation to this dataset in that the dataset only accounts for Android users and those who always leave their device GPS on.

We utilized several methods to determine lag days that will be used in this study. First, we seek evidence available online on COVID-19 incubation periods until the infected persons develop symptoms. We assumed that people would go for COVID-19 tests after developing symptoms, which is around 7–14 days [17,18]. Then, we use a cross-correlation function to explore correlation for each lag day between daily confirmed cases and mobility category. To ascertain the lag days, we will incorporate several lag days, according to cross-correlation function results into several models and compare it between models based on Akaike Information Criterion (AIC), and Root Mean Square Error (RMSE). Regression models were used to predict the continuous dependent variable, with COVID-19 daily confirmed cases as the dependent variable in this study. Further, 7-Day Moving Average (7DMA) was applied to the COVID-19 case data to reduce the weekend–weekday bias of data counting. We found a skewed distribution in the COVID-19 case data following the analysis. Therefore, we conducted log-transformation on the data to follow the normal distribution to fit our model better. We are aware that when the data are log-transformed, it is significant to demonstrate the interpretation concerning the original data [19,20,21].

We explored several models to estimate COVID-19 cases. We included Poisson Regression GLM and Negative Binomial GLM in the analysis as both analyses were suitable for modeling count data. We also added Multiple Linear Regression (MLR) for the comparison. Therefore, we compared three models: Model 1. Poisson Regression GLM, Model 2. Negative Binomial Regression GLM and Model 3. Multi Linear Regression. We used parameters of Akaike Information Criterion (AIC) and Root Mean Square Error (RMSE) to assess the best fit model. A larger R-Square value indicates the model better fits the observations. Meanwhile, the lower values of AIC and RMSE indicate a better-fit model.

### 2.3. Variable Selection

The mobility types recorded in Google CMR are not independent of each other, e.g., people might go to workplaces, groceries, and public transit in a day. Hence, multicollinearity of the data exists. However, having different combinations of mobility gives different levels of exposure. Therefore, we applied Principal Component Analysis (PCA) to consider the contribution of different types of mobility to the overall mobility exposures.

## 3. Results

We used a cross-correlation function to determine lag days applied to the dataset. This function applied Pearson Correlation to the independent variables while applying lag simultaneously. The result of cross-correlation function analysis can be seen in Figure 1. Note that three variables have a higher correlation on positive lag, meaning notable increases in daily confirmed cases, happening after 6 to 20 days of increased mobility on variables “Grocery and Pharmacy”, “Retail and Recreation”, and “Workplaces”. Detailed number on cross-correlation funciton value are available in Appendix A.

Then, we continue with the results from the cross-correlation test with a regression model. Combined with assumptions that COVID-19 symptoms develop within 7–14 days, we took seven days as lag applied to the variables. Table 1 shows a comparison between lag days tested on “Grocery and Pharmacy” mobility. Based on those results, we note that 7-day lags have smaller AIC that we take account as the lag days used in this model.

Based on the R-Square values, we found that the MLR model, where all mobility variables were included, explained the variance by 66.39%. Furthermore, given the analysis result, it further showed that groceries and pharmacies, transit stations, and workplaces indicate some multicollinearity shown by the coefficient (Table 1). A positive coefficient was shown when variables were inserted individually but was then affected by other independent variables in the multi-linear regression analysis.

We applied Principal Component Analysis (PCA) to develop a single composite variable to address the multicollinearity. Three categories that showed multicollinearity (grocery and pharmacies, transits stations, and workplaces) were combined into one composite index using a z-score. To this end, we created five models, which consisted of different predictors: (1) parks and retail and recreation; (2) parks, retail and recreation, and groceries; (3) parks, retail and recreation, and transit; (4) parks, retail and recreation, and workplaces; and (5) parks, retail and recreation, and one single index of groceries, transit stations and workplaces.

As seen in Table 2, Multiple Linear Regression showed a better overall result in lower AIC and RMSE than Poisson Regression and Negative Binomial Regression. When comparing the first four models with the last model where PCA was applied, the latter showed better results with a lower AIC value (2.29) and RMSE value (0.76), making it the most suitable model that could provide a better explanation. This model also explained the variance in COVID-19 cases by 52%. Hence, we chose this model to be further analyzed in this study.

Multiple Linear Regression analysis was then conducted to explore the influence of each mobility variable on COVID-19 cases. In this step, the predictors were (1) single z-score index of grocery, transit, workplaces; (2) parks; and (3) retail and recreation.

Based on the result of the Multiple Linear Regression analysis in Table 3, the equation is as follows:log (7-day MA daily confirmed case) = 8·44 + 0.41 × (Z_Score Grocery Transits Workplaces) + 0.02 × (Parks) + 0·03 × (Retail and recreation)

Because, previously, we applied log transformation of the COVID-19 daily confirmed cases and used a composite variable, further steps were needed to interpret the overall results to observe the influence of each mobility category on daily confirmed cases. First, the combined variable coefficient needs to be divided with its respective standard deviation (Table 2) as the denominator. Subsequently, the division result needs to be exponentiated (Table 4). The same equation needed to be performed to the confidence interval. Meanwhile, the coefficient of the other separated two mobility variables (parks, retail and recreation) only need to be exponentiated.

Accordingly, we found that “grocery and pharmacies” contributed the highest percentage among all mobility variables. A one percent increase in “grocery and pharmacies” mobility contributed to a 4.12% increase in cases. Retail and recreation mobility contributed to a 3.11% increase. Transit stations and workplaces contributed to the rise in cases by 2.26% and 2.56%, respectively. Meanwhile, parks accounted for a 1.93% rise in the cases (Table 5).

### COVID-19 Cases and Containment in Jakarta

As seen in Figure 2, the mobility changes in population mobility trends are in line with the existing restriction policy and regulations and the holidays. The government revised Jakarta’s mobility restrictions several times from 15 February until 31 December 2020, adjusting for the dynamics. The regulation includes large-scale social restrictions (PSBB), followed by a relaxation (PSBB Transisi), shifting back to the tighter restrictions (PSBB), and back to a relaxation (PSBB Transisi and PSBB Transisi Extended), as the cases seemed to be dropping down. It can also be observed from the figure that religious holidays (Eid Al-Fitr) and public holidays, where some lead to long weekends, caused community mobility changes and increased COVID-19 cases.

Additionally, where the observed (actual cases) and predicted values of COVID-19 new cases based on the model were shown, a gap in the period of middle-to-late September towards late November appeared during PSBB. We assumed that this gap occurred due to the nature of the data. We observed that almost all of the mobility categories showed a decline in trends in the same period (Figure 3). Nevertheless, to be noted, this model does not use time-dependent data. Therefore, when the mobility decline is found in the period starting from the middle of September (PSBB), the model estimated a reduction in cases.

## 4. Discussion

### 4.1. Effect of Mobility by Categories on COVID-19 Dynamics

This study explored the utilization of mobility data to predict COVID-19 dynamics in Jakarta. This study observed that different types of community mobility partially explained the COVID-19 dynamics in Jakarta in 2020. All mobility categories (“groceries and pharmacy”, “retail and recreation”, “transit stations”, “workplaces”, and “parks”) were significant predictors of COVID-19 cases. Our study suggests that mobility data provided by publicly available datasets are reliable to be used in decision making and policy formulation. Local governments can use this model to improve health system readiness by anticipating case surge.

The models we used in this study demonstrated the relationship between mobility and COVID-19 dynamics with the fewest assumptions possible, as CMR data were some of the most feasible publicly available data at the beginning of the pandemic. We further intended to explore the impact that community movement would have on COVID-19 cases and identify the places with a high contribution towards a case increase. After several statistical model explorations, we observed that simple statistics through Multiple Linear Regression could be utilized to explain the impact. However, as the mobility categories recorded in the Google CMR are not independent of each other, e.g., people might go to workplaces, groceries, and public transit in a day, MLR assumptions of no multicollinearity did not meet. Therefore, variable adjustments through Principal Component Analysis (PCA) were applied to consider different types of mobility to the overall mobility exposures. In addition, some confounding and other possible limitations existed during the data exploration. Thus, the results in this study should be interpreted with caution.

Mobility on groceries and pharmacies contributed the highest increase in COVID-19 new cases. This finding is in line with a previous study in India, where travel for daily need purposes were related to COVID-19 transmission [22]. During the restrictions, food warehouses, farmers’ markets, specialty markets, and drugstores were among the essential businesses allowed to open. People shifted from restaurants towards groceries and food sellers during the stay-at-home orders [23]. The COVID-19 Task Force revealed around 107 traditional market clusters, and 555 cases were reported from June to July 2020 [24]. In China, the first wave of COVID-19 was identified in a seafood market in Wuhan. Later in June, the Chinese government reported 57 new cases resulting from the new transmission cluster in one market area in Beijing [25].

A report from the COVID-19 Task Force noted that traditional markets contributed 4.3% of COVID-19 new cases in Jakarta [26]. Since the beginning of the pandemic, the government began enacting a massive public campaign on health protocols, namely mask wearing, hand washing, and physical distancing. The coverage of this protocol implementation was required in many areas, including traditional markets. However, traditional markets tend to be crowded and unorganized, making implementation and public compliance challenging. Data reported in September 2020 showed that compliance towards health protocols in traditional markets is 50.6%, making it the lowest compared to workplaces (86%) and shopping malls/plazas (80.71%) [27]. For that reason, we suggest that closer attention should be given to traditional markets, including solid implementation of health protocols, followed by adequate testing and tracing surveillance, anticipating case surges and curbing the transmission to other places.

Various places, including cafés, restaurants, shopping malls, recreation areas, and parks, in Jakarta were shut down in March–June and reopened, adjusting around 50% of the total capacity during the first relaxation period, triggering high mobility to those places. Earlier studies found that virus transmission most likely occurred in indoor environments, particularly in crowded and poorly ventilated areas [28]. Therefore, without interpersonal distance and ventilation improvement, there is a strong possibility of the airborne transmission of COVID-19 in the retail and recreation sectors [29].

At the beginning of the pandemic, the government of Jakarta implemented tight restrictions by limiting the number of passengers on public transportation by implementing the SIKM (Exit–Entry Access) from and to Jakarta to reduce transmission within the city and to other cities/provinces. However, as the restrictions were relaxed and religious and public holidays went ahead, the visits to transit stations within the city increased. Confined spaces and closed environments on public transportation have a high risk of infectious disease transmission [30], since it potentially leads to crowding. As for workplaces, the distinction of mobility was not very clearly observed. Some people were still working in the informal sectors, and some office workers were still required to work from the office, thus, explaining the minor fluctuation trends in this sector. Workplaces were places with better compliance to health protocols, notably in mask wearing (94.83%), temperature check (94.35%), and washing hands (82.56%) [31].

### 4.2. Mobility Relaxation, Seasonal Events, and COVID-19 Dynamics

There are many public holidays throughout the year, including religious holidays. Some of the public holidays coincidentally lead to long weekends. One of the biggest religious events, Eid-ul Fitr, has been predicted to cause a population mobility spike due to annual migrations, known as homecoming (mudik). The government enforced travel restrictions to anticipate a significant surge in COVID-19 cases due to high levels of travel, yet we observed a COVID-19 case increase a few weeks after the event. We assume that despite the restrictions, mobility was still high within the city, such as visiting retail and recreation, which still allowed up to 50% total capacity. Essential sectors, including groceries and pharmacies, were excluded from the restrictions. In addition, the capital city, Jakarta, is home to a large number of migrants. We argue that the resistance towards the travel restrictions might be high as some would not want to miss the once-in-a-year homecoming event. Therefore, movement in and out of the city was allowed. Airports and train stations also remained in operation during the restrictions.

Besides, low adherence to health protocols may exacerbate the case surge. In the last quarter of 2020, Indonesian COVID-19 task force data reported many people who neglected health protocols in Jakarta during the holiday. On 27 November 2020, the report showed that the compliance towards mask wearing and physical distancing was 58.32% and 43.46%, respectively. This finding suggests that long holidays might trigger a decline in health protocol compliance [32]. Hence, we suggest that mobility restrictions are not the only key driver for the case surge after long holidays.

Prolonged pandemic situations and mobility relaxation might influence lower adherence towards mobility and health protocols. After the first relaxation period, the mobility trend in Jakarta rose gradually. When the government of Jakarta implemented the second restriction in September, a reduction in mobility was observed and then increased again after the relaxation was put into effect in October. A study in the United States on the relationship of human mobility and COVID-19 infections demonstrated that after stay-at-home orders in a state of emergency in the early pandemic, mobility decreased around 35%. However, as the announcement of the partial reopening and the signs of pandemic fatigue have shown, mobility rebounded rapidly, leading to a positive relationship in COVID-19 infections [33].

Mobility relaxations and the news regarding the development of COVID-19 vaccines approaching the end of the year might bring a glimpse of hope that might cause risk compensation. Though the long-term effect of the vaccines was still unclear at that time, people could still see this as a silver lining. This situation was previously explained as the Peltzman Effect, where individuals respond to safety measures while increasing their risky behavior [34]. The vaccine development might suggest that the pandemic is getting somewhat closer to the end. This situation led people to compromise the preventive measures and resulted in risky behavior, such as lower compliance to stay at home and for social distancing [34,35].

The limitations of this study are related to the study’s nature, where the analysis using CMR could not solely be the primary evidence to explain COVID-19 dynamics. The information on CMR only captures the mobility information from those who have access to smartphones and turn on their location settings. In addition, recent mobility information on Google CMR represents data from several days prior, making it challenging to utilize for present decision-making purposes.

Our study utilized only community mobility data to explore the effect of mobility on COVID-19 dynamics, resulting in a partial explanation of the dynamics. Previous studies suggested that compared to other behavioral changes (wearing face masks, hand washing, social distancing), mobility played a minor role in COVID-19 transmission. Travel restrictions played a minor role in the spread [36,37]. In Asia, South Korea never implemented a lockdown policy but still managed to have no transmission cases during the election period by emphasizing strict compliance towards health protocols [38].

Therefore, we note that exploration of other drivers of transmission, in addition to mobility, is necessary. Those contributing factors may include the nature of the virus, public health interventions, and individual behavior [39,40]. Exploration of indoor and outdoor crowds is also necessary to enhance the analysis, as CMR data do not provide such information. This study captured the situation at the city level, representing a within-city movement of the population. Many modes of transportation are available in Jakarta; inter-city movement can be taken into account to explain COVID-19 dynamics further. We suggest that future studies use a more dynamic approach through agent-based modeling simulation to get more comprehensive information on various factors contributing to COVID-19 dynamics. Further, a combination with daily confirmed case data when a mobility restriction period is applied can be used to gain further insight into mobility effects for COVID-19 cases.

## 5. Conclusions

This study demonstrated that mobility data obtained from CMR could be utilized to predict COVID-19 dynamics, shown by the different impact of each mobility category on COVID-19 dynamics in Jakarta. Mobility management and adherence to health protocol are critical during relaxations and public holidays where mobility is expected to increase. Epidemic management, such as testing and tracing, should focus on public places, which potentially contribute to increased COVID-19 cases.

This study also showed that any available data, though imperfect, could be utilized to provide a basis for timely intervention and control, as well as immediate decision making and policy formulation at a time of public health crisis, particularly at a time when the available data are limited. Decision makers in public health could utilize the current CMR data analysis with basic statistical modeling to anticipate increased COVID-19 cases, especially during seasonal events, such as annual religious holidays or other long holidays.

## Figures and Tables

**Figure 1 ijerph-19-06671-f001:**
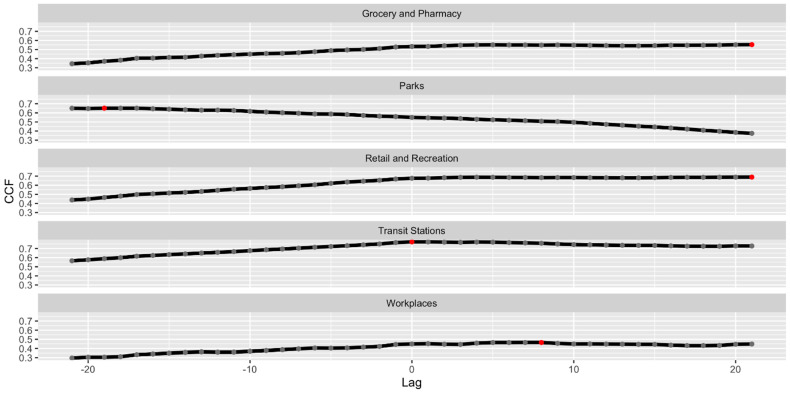
Correlation plot between independent variables and daily confirmed case variables at a different time lag.

**Figure 2 ijerph-19-06671-f002:**
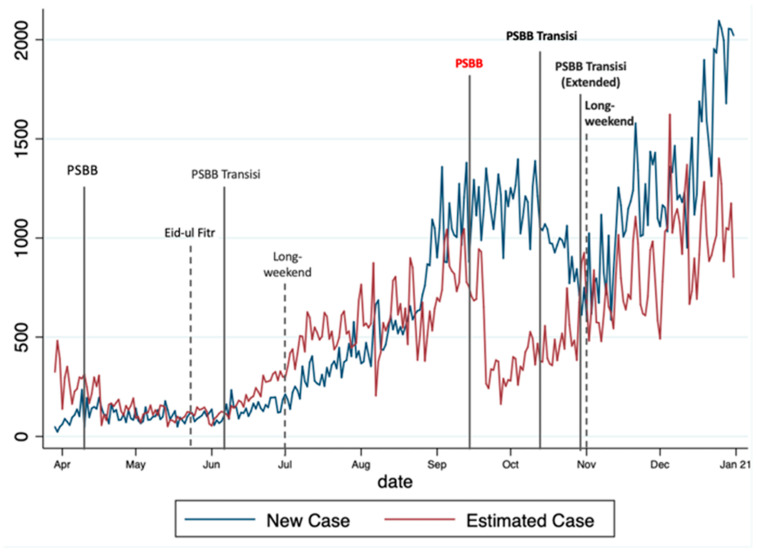
Observed and estimated values of COVID-19 cases.

**Figure 3 ijerph-19-06671-f003:**
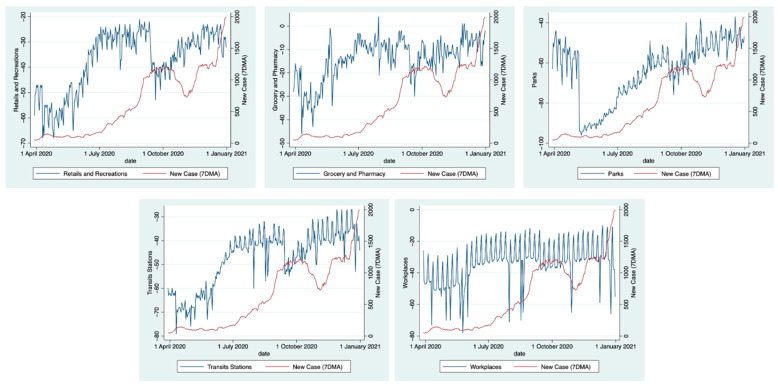
Changes in human mobility in Jakarta, 2020.

**Table 1 ijerph-19-06671-t001:** Comparison between lag days based on AIC, RMSE, and R-squared.

Lags	AIC	RMSE	R^2^
7 days	757.02	0.92	0.28
14 days	793.28	0.98	0.18

**Table 2 ijerph-19-06671-t002:** Multiple linear regression analysis of the mobility categories and new COVID-19 daily confirmed cases.

Variables	Coefficient	Std. Error	*p* > |t|	[95% CI]
Cons.	10.25	0.21	0.00	(9.84, 10.67)
Retail and recreation	−0.01	0.01	0.08	(−0.03, 0.00)
Grocery and Pharmacies	−0.04	0.01	0.00	(−0.06, −0.01)
Parks	0.00	0.00	0.15	(−0.00, 0.01)
Transits Stations	0.14	0.01	0.00	(0.11, 0.16)
Workplaces	−0.04	0.01	0.00	(−0.05, −0.33)

Daily confirmed case variable is in 7-day moving average, log-transformed; mobility variables are lagged seven days.

**Table 3 ijerph-19-06671-t003:** Poisson GLM, Negative Binomial GLM, and Multiple Linear Regression of COVID-19 Daily Confirmed Cases.

Model	AIC			RMSE		
	Pois	NB	MLR	Pois	NB	MLR
1. Parks_Retails	3.75	5.73	2.33	0.78	0.78	0.77
2. Parks_Retails _Grocery	3.76	5.74	2.34	0.78	0.78	0.77
3. Parks_Retails_Transits	3.74	5.74	2.15	0.71	0.71	0.70
4. Parks_Retails_Workplaces	3.76	5.74	2.31	0.77	0.77	0.76
5. Parks_Retails_Z-Score Grocery_Transits_Workplaces	3.76	5.74	2.29	0.76	0.76	0.76

Daily confirmed cases are log-transformed for the Multi-Linear Regression analysis.

**Table 4 ijerph-19-06671-t004:** Multiple linear regression analysis using daily confirmed case and mobility variables.

Variables	Coef.	Std. Err.	*p* > |t|	95% CI
Cons.	8.44	0.28	0.00	(7.88, 9.00)
Z_Score_Grocery_Transits_Workplaces (Lagged 7 days)	0.41	0.12	0.00	(0.18, 0.65)
Parks (Lagged 7 days)	0.02	0.00	0.00	(0.01, 0.02)
Retail and recreation (Lagged 7 days)	0.03	0.00	0.00	(0.01, 0.45)

**Table 5 ijerph-19-06671-t005:** The regression coefficient of mobility variables.

Variable	Coef. (Exp)	Coef. (%)	Mean	Std. Dev	95% Confidence Interval (Exp)
Grocery and pharmacy	1.04	4.12	−12.30	10.34	(1.01, 1.06)
Transits stations	1.02	2.26	−42.98	18.64	(1.00, 1.03)
Workplaces	1.02	2.56	30.01	16.54	(1.01, 1.04)
Parks	1.01	1.93	−57.22	24.02	(1.01, 1.02)
Retails and recreation	1.03	3.11	−34.03	16.03	(1.01, 1.04)

## Data Availability

Google Community Mobility Reports Data can be accessed via https://www.google.com/covid19/mobility/ (accessed on 10 January 2021). COVID-19 daily confirmed case data can be accessed through https://kawalcovid19.id/ (accessed on 10 January 2021).

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
