# Peer review of "Community Mobility and COVID-19 Dynamics in Jakarta, Indonesia"

_ijerph, 2022, doi:10.3390/ijerph19116671_

Round 1
Reviewer 1 Report
Referee report for “Community Mobility and COVID-19 Dynamics in Jakarta, Indonesia”; Manuscript ID: ijerph-1696397, for IJERPH
This study examines the effect of human mobility (measured by the Google community mobility reports) on the COVID-19 incidence of an urban population in Jakarta, Indonesia.
Although the research is timely, I found the contribution very limited, especially for an International journal and audience. Moreover, several problems exist in the analysis of the data. Detailed review follows.
Major comments
From lines 79-81, it appears that the purpose of the study is etiological but the methods used are very descriptive and focus on prediction rather than causality.
Regarding the lag selection, I am afraid that just mentioning the command of statistical software does not clarify the method used and how the lag was chosen (line 118). Nevertheless, from what I understand, choosing based on the highest value of a cross correlation is not a sound way of choosing. Instead, one can use the Schwarz's Bayesian information criterion (SBIC), the Akaike's information criterion (AIC), or the Hannan and Quinn information criterion (HQIC). Since authors use STATA I invite them to take a look at the -varsoc- command. I would rely on AIC for the particular data if there is disagreement. One can also choose to include more than one lag if the sole aim is prediction.
There is no need to transform the cases data for analysis with Poison or NB models. The problem of these models is that observations are not independent due to serial correlation, hence the likelihood that these models maximise is incorrect. There is no mention on whether the standard errors of the linear model were corrected for heteroskedasticity, since errors are definitely not homoskedastic for the same reason as before. Lastly, I don’t know how the authors obtained a comparable to OLS R squared from ML estimators, since such thing does not exist.
No details were given for the parameters of the PCA. Why were multiple PCA performed from indices? One PCA using all indices would provide a single component that would do the job.
A descriptive statistics table is missing, along with all the information that it should contain.
lines 156-7. A simple Pearson correlation is most of the times enough to detected serious collinearity. In a model setting, one can calculate the variance inflation index (-vif- after linear estimation command). I do not understand how authors concluded multicollinearity from the p-values of a regression model.
lines 187-194. I highly doubt that you can decompose a variable created by PCA after the linear transformation in the way described by the authors. Where is that method described in detail?
Minor comments
According to line 84, this is a cross-sectional study but the title talks of COVID-19 dynamics. On the other hand, it analyzes a time series of cases. This is confusing.
lines 196-7. I don’t think it is necessary to explain this for an academic audience.
Figure 2 is too small. Unreadable.
line 238. Not including any control variables and ignoring the dependence of the data is the opposite of “…the fewest assumptions possible..”.
line 248. Definitely exist.
line 328. I believe that by now it is clear that there is no herd immunity for this virus.
lines 360-1. This was not investigated in this study, so it is just the authors’ opinion.
Some links would be nice for the non-article citations.
Author Response
Response to Reviewer 1
Major comments
From lines 79-81, it appears that the purpose of the study is etiological but the methods used are very descriptive and focus on prediction rather than causality.
Response: Thank you for highlighting this issue. Our study focused on building predictive model for COVID-19 dynamics using statistical modeling techniques. Our question was whether the mobility data could be used to predict COVID-19 dynamics as during the early COVID-19 pandemic there were only a few available data that could be used for decision making. As such, Google mobility data was readily available data on dynamics of mobility in Indonesia.
Regarding the lag selection, I am afraid that just mentioning the command of statistical software does not clarify the method used and how the lag was chosen (line 118). Nevertheless, from what I understand, choosing based on the highest value of a cross correlation is not a sound way of choosing. Instead, one can use the Schwarz's Bayesian information criterion (SBIC), the Akaike's information criterion (AIC), or the Hannan and Quinn information criterion (HQIC). Since authors use STATA I invite them to take a look at the -varsoc- command. I would rely on AIC for the particular data if there is disagreement. One can also choose to include more than one lag if the sole aim is prediction.
Response: Thank you for your comment on this issue. Your suggestion of using AIC was incorporated in our study to determine the lag days and added the explanation in the manuscript as follows: “We utilized several methods to determine lag days that will be used in this study. First, we seek for evidence available online on COVID-19 incubation periods until the infected persons develop symptoms. We assumed that people will go for COVID-19 test after developing symptoms, which is around 7-14 days [20, 21]. Then, we use a cross-correlation function to explore correlation for each lag day between daily confirmed case and mobility category. To ascertain the lag days, we incorporated several lag days according to cross-correlation function results into several models and compared it between models based on Akaike Information Criterion (AIC), and Root Mean Square Error (RMSE).”
There is no need to transform the cases data for analysis with Poison or NB models. The problem of these models is that observations are not independent due to serial correlation, hence the likelihood that these models maximise is incorrect. There is no mention on whether the standard errors of the linear model were corrected for heteroskedasticity, since errors are definitely not homoskedastic for the same reason as before.
Response: Thank you for outlining this issue. We only apply log transformation on multiple linear regression model since one of the assumptions need to be met is to avoid the heteroskedascity. With log-transformation to the dependenet variable in multiple linear regression model, the result will correct the heteroskedascity.
Lastly, I don’t know how the authors obtained a comparable to OLS R squared from ML estimators, since such thing does not exist.No details were given for the parameters of the PCA. Why were multiple PCA performed from indices? One PCA using all indices would provide a single component that would do the job. A descriptive statistics table is missing, along with all the information that it should contain. lines 156-7. A simple Pearson correlation is most of the times enough to detected serious collinearity. In a model setting, one can calculate the variance inflation index (-vif- after linear estimation command). I do not understand how authors concluded multicollinearity from the p-values of a regression model. lines 187-194. I highly doubt that you can decompose a variable created by PCA after the linear transformation in the way described by the authors. Where is that method described in detail?
Response: Thank you for pointing out this issue. The reason we performed multiple PCA from the variables was due to the interdependency of each variable with one another. Based on the coefficient of the initial analysis, we observed that if each of the variable was inserted individually, it will result in multicollinearity (observed in three out of five variables). Hence, combining them into one single index with PCA is necessary. In terms of detecting the multicollinearity, we would like to clarify that we actually concluded multicollinearity based on the coefficient result–not p-value, (we apologize for the miswritten in the manuscript, change has been made). As an addition in determining the multicollinearity, when every variable inserted individually, the coefficients should have given positive results, but it was affected by other independent variables (coefficients become negative). The variables in community mobility runs interdependently, and visit to workplaces, groceries, and transits altogether may increase the risk exposure.
Minor comments
According to line 84, this is a cross-sectional study but the title talks of COVID-19 dynamics. On the other hand, it analyzes a time series of cases. This is confusing.
Response: We sincerely thank you for highlighting this issue. We would like to clarify that we obtained the daily cross-sectional data to be analyzed in the time-series frame. As mentioned previously, we only have access to currently available data (daily aggregate confirmed COVID-19 case and mobility data), which aren’t suitable for causal inference analysis. This study demonstrated that any available existing data, even with the limitations and other imperfections, could be utilized to support immediate policy-making, especially in the time of public health crisis. We changed the wordings to explain this in line 98-9.
lines 196-7. I don’t think it is necessary to explain this for an academic audience.
Figure 2 is too small. Unreadable.
Response: Thank you for the inputs. Changes have been incorporated in the manuscript. As for the figure, the previous one was exchanged with a bigger and clear one. We changed the order of the figure, the figure is now Fig. 3 (Line 294).
line 238. Not including any control variables and ignoring the dependence of the data is the opposite of “…the fewest assumptions possible..”.
Response: Thank you for raising your concern in this issue. In this study, we attempted to make a prediction by making the most out of existing available data. Google Mobility has provided publicly available data on mobility categories in the early phase of the pandemic. Therefore, we decided to first try to utilize the mobility data as a parsimonious model.
line 248. Definitely exist.
line 328. I believe that by now it is clear that there is no herd immunity for this virus.
lines 360-1. This was not investigated in this study, so it is just the authors’ opinion.
Some links would be nice for the non-article citations.
Response: Thank you for the inputs and we have incorporated it in the manuscript. For the line 248 we changed the sentence in line 326-67. Please find the links for the non-article citation highlighted in yellow in the manuscript.
Reviewer 2 Report
- Limitations of Google CMR should be discussed in the methodology section i.e. penetration, and others.
- programming and policy implication of this study should be embolden in the work especially in the discussion and conclusion section.
- A follow-up study should combine the CMR if possible with data on COVID-19 restrictions i.e. social distance, face mask, handwashing etc., and control for them if possible.

Author Response
Response to Reviewer 2
- Limitations of Google CMR should be discussed in the methodology section i.e. penetration, and others.
Response: Thank you for your comment in this issue. We already add the limitation of the Google Mobility Dataset in the Data on Community Mobility section.
- Programming and policy implication of this study should be embolden in the work especially in the discussion and conclusion section.
Response: Thank you for your suggestion. We already add potential policy implications in the Discussion and Conclusion section.
- A follow-up study should combine the CMR if possible with data on COVID-19 restrictions i.e. social distance, face mask, handwashing etc., and control for them if possible.
Response: Thank you for your suggestion. We add more on further study suggestion in the last paragraph of Discussion section.
Reviewer 3 Report
The manuscript consists of total 11 pages, including 4 tables, 3 figures and the list of total 37 literature references. The article presents original results of the study that aimed at confirming links between the population mobility changes estimated by using the cellular phone location services data and the COVID-19 pandemic dynamics; it is based on the sound theory that increased population mobility shall result in an increase in the number of social interactions of the infected individuals and thus also in increased risk of passing the infection to others. The topic has the practical value in providing to the skeptics the confirmation of the rationale and justification for shut-downs and other limitations imposed by governments on the citizens during the COVID-19 pandemic. As such the topic of the article fits into the scope of works published by the Journal. The article is written in good quality English.
The title of the manuscript is adequate to its contents.
The Abstract mirrors both the structure and the main contents of the main text adequately.
The Introduction provides broad enough background information for the explored scientific problem.
The Material and Methods section is describing the accepted methodology of the study in high detail.
However, there is a quite serious matter that may have distorted the Authors' findings and their interpretation: They assumed (line 147) that the COVID-19 symptoms demonstrated within 7-14 days after the infection has been acquired while in fact the average SARS-CoV-2 infection incubation period was significantly shorter, it was about 5 days (most often from 4 to 6 days but in case of some individuals it was ranging from 1 to even 14 days). The Authors may consider recalculating their results to accommodate the fact above or pointing in the article on the evidence supporting the thesis that in the particular population They studied the incubation period was indeed longer than on average reported elsewhere.
The Results section is consistent with the methodology accepted by the Authors, richly supported by tables and figures.
The Discussion section places the Authors findings in the context of the published information published by other authors.
The Conclusion is based on the discussed results.
The References are numerous and recent enough, relevant to the topic of the article.
The Authors may enrich the background and context of the presented problem by mentioning the following aspects of COVID-19 pandemic:
- mentioning the other than Google's means of population mobility and contacts exploration and its uses during the COVID-19 pandemic, as e.c. in: https://doi.org/10.3390/ijerph19074383 https://doi.org/10.3390/su14074248 https://doi.org/10.3390/ijerph19053129 https://doi.org/10.3390/ijerph19031410 https://doi.org/10.3390/ijgi11010060 https://doi.org/10.3390/economies9040182
- mentioning the characteristics of government interventions impact on the population mobility and its results, as e.c. in: https://doi.org/10.3390/su14063694 https://doi.org/10.3390/su14063567 https://doi.org/10.3390/land11030429 https://doi.org/10.3390/su14063368 https://doi.org/10.3390/ijerph182312567
Author Response
Response to Reviewer 3
The manuscript consists of total 11 pages, including 4 tables, 3 figures and the list of total 37 literature references. The article presents original results of the study that aimed at confirming links between the population mobility changes estimated by using the cellular phone location services data and the COVID-19 pandemic dynamics; it is based on the sound theory that increased population mobility shall result in an increase in the number of social interactions of the infected individuals and thus also in increased risk of passing the infection to others. The topic has the practical value in providing to the skeptics the confirmation of the rationale and justification for shut-downs and other limitations imposed by governments on the citizens during the COVID-19 pandemic. As such the topic of the article fits into the scope of works published by the Journal. The article is written in good quality English.
The title of the manuscript is adequate to its contents.
The Abstract mirrors both the structure and the main contents of the main text adequately.
The Introduction provides broad enough background information for the explored scientific problem.
The Material and Methods section is describing the accepted methodology of the study in high detail.
However, there is a quite serious matter that may have distorted the Authors' findings and their interpretation: They assumed (line 147) that the COVID-19 symptoms demonstrated within 7-14 days after the infection has been acquired while in fact the average SARS-CoV-2 infection incubation period was significantly shorter, it was about 5 days (most often from 4 to 6 days but in case of some individuals it was ranging from 1 to even 14 days). The Authors may consider recalculating their results to accommodate the fact above or pointing in the article on the evidence supporting the thesis that in the particular population They studied the incubation period was indeed longer than on average reported elsewhere.
The Results section is consistent with the methodology accepted by the Authors, richly supported by tables and figures.
The Discussion section places the Authors findings in the context of the published information published by other authors.
The Conclusion is based on the discussed results.
The References are numerous and recent enough, relevant to the topic of the article.
The Authors may enrich the background and context of the presented problem by mentioning the following aspects of COVID-19 pandemic:
- mentioning the other than Google's means of population mobility and contacts exploration and its uses during the COVID-19 pandemic, as e.c. in: https://doi.org/10.3390/ijerph19074383 https://doi.org/10.3390/su14074248 https://doi.org/10.3390/ijerph19053129 https://doi.org/10.3390/ijerph19031410 https://doi.org/10.3390/ijgi11010060 https://doi.org/10.3390/economies9040182
- mentioning the characteristics of government interventions impact on the population mobility and its results, as e.c. in: https://doi.org/10.3390/su14063694 https://doi.org/10.3390/su14063567 https://doi.org/10.3390/land11030429 https://doi.org/10.3390/su14063368 https://doi.org/10.3390/ijerph182312567
Response: We sincerely appreciate your comments and valuable inputs into our manuscript. About the COVID-19 incubation period, we learned that during that time the longest incubation of SARS-CoV-2 is 14 days and the government (through the ministerial decree in 2020) stated that the large-scale social restriction (PSBB) was applied based on the longest incubation period of 14 days and this regulation affected Jakarta as the epicenter of the pandemic in Indonesia at that time. Therefore, we took the 7-14 days range of incubation period in this study. Some other studies also mentioned the 7-14 days incubation period in Indonesia at that time.
Reference:
- https://www.ncbi.nlm.nih.gov/pmc/articles/PMC7105032/
- Rachman, B.E. & Rusli, Musofa & Miftahussurur, Muhammad. (2020). The hidden vulnerability of COVID-19 observed from asymptomatic cases in Indonesia. Systematic Reviews in Pharmacy. 11. 703-713. 10.31838/srp.2020.2.103.: https://doi.org/10.15294/ijicle.v2i2.38324
- https://apps.who.int/iris/rest/bitstreams/1369903/retrieve
- http://berkas.dpr.go.id/puslit/files/info_singkat/Info%20Singkat-XII-3-I-P3DI-Februari-2020-1957-EN.pdf
Furthermore, thank you for your valuable suggestion on enrich the background and context of our study. We have incorporated this suggestion in our manuscript by adding some of the articles in the background (line 57-9; 72-7).
Reviewer 4 Report
This study presents informative inputs regarding COVID-19. It illustrates a partial explanation of COVID-19 dynamics in Indonesia. In addition, good concepts are added to the discussion, such as the "Peltzman Effect".
Line 55-56: consider deleting “reductions”
We use multiple linear regression analysis when our dependent variable is either continuous, a scale variable, or interval level. However, it can be carefully used to analyse discrete data, it would be good if justify it in “methodology”. Does “log transformation of the COVID-19 daily cases” mean for that? Alternatively, describe it in the limitation of the study.

Author Response
Response to Reviewer 4
This study presents informative inputs regarding COVID-19. It illustrates a partial explanation of COVID-19 dynamics in Indonesia. In addition, good concepts are added to the discussion, such as the "Peltzman Effect".
Line 55-56: consider deleting “reductions”
We use multiple linear regression analysis when our dependent variable is either continuous, a scale variable, or interval level. However, it can be carefully used to analyse discrete data, it would be good if justify it in “methodology”. Does “log transformation of the COVID-19 daily cases” mean for that? Alternatively, describe it in the limitation of the study.
Response Thank you for your valuable comments, suggestions, and highlight on the issue of linear regression. As suggested, we have removed the word ‘reductions’ in the manuscript. As for the linear regression analysis, we are aware that the original data is discrete. But the data was subsequently smoothed using moving average to make it non-discrete, making it reasonable to use the log transformation in the linear model to analyze non-discrete data. Hence, we do not think that the log-transformation as a limitation of our study, as this study attempted to utilize available data, though imperfect, to provide a basis for timely intervention and control as well as immediate decision-making in the time of public health crisis, when there is a limited data available.
Round 2
Reviewer 1 Report
The revised version is an improvement. I invite the authors to give more emphasis to lag selection, unit roots, cointergration, granger causality and testing for structural breaks and serial correlation when they analyse time-series data. All these are important since time series are never smooth functions that can be easily approximated and special models are needed to account for these issues and the additional complexity.
Author Response
The revised version is an improvement. I invite the authors to give more emphasis to lag selection, unit roots, cointergration, granger causality and testing for structural breaks and serial correlation when they analyse time-series data. All these are important since time series are never smooth functions that can be easily approximated and special models are needed to account for these issues and the additional complexity.
Respond: Thank you for your positive review and for highlighting this important issue. We agree that the issues that you mentioned are important to establish evidence of causality for time series analysis between population mobility and the transmission dynamic of Covid 19. However, we only conducted descriptive comparison as we assumed that mobility was not the only determinant for Covid 19 transmission.
Furthermore, we did not be able to control other determinants (e.g., social interaction, individual risk prevention) into our model. Our descriptive comparison between the predicted and observed cases support our assumption. As you can see in Figure 2, our model shows a good fit with the observed data before October 2020. However, when the government enforce “lockdown” mid-September 2020 (PSBB), our model predicted a direct significant drop in number of cases while the daily reported cases data showed only a slight reduction of new cases. This indicates that a significant drop in mobility does not directly cause a significant drop in transmission, as a high transmission might still happened in the household setting.
Additionally, as previously mentioned, our model did not intend to find causality between two time series variables. However, event with this study limitation, we believe that the study is still relevant for policy makers to help them to be able to make in time appropriate mobility control measure.
